# Risk Assessment of Isoeugenol in Food Based on Benchmark Dose—Response Modeling

**DOI:** 10.3390/toxics11120991

**Published:** 2023-12-05

**Authors:** Thomas Quentin, Heike Franke, Dirk W. Lachenmeier

**Affiliations:** 1Postgraduate Study of Toxicology and Environmental Protection, Rudolf-Boehm-Institut für Pharmakologie und Toxikologie, Universität Leipzig, Härtelstrasse 16–18, 04107 Leipzig, Germany; tquentin@gmx.de (T.Q.); heike.franke@medizin.uni-leipzig.de (H.F.); 2Chemisches und Veterinäruntersuchungsamt (CVUA) Karlsruhe, Weissenburger Strasse 3, 76187 Karlsruhe, Germany

**Keywords:** benchmark dose, isoeugenol, acceptable daily intake, cancer risk

## Abstract

Isoeugenol has recently been evaluated as possibly carcinogenic (Group 2B) by the WHO International Agency for Research on Cancer (IARC). In light of this evaluation, an updated risk assessment of this common food constituent was conducted using the benchmark dose (BMD) approach as recommended by the European Food Safety Authority (EFSA) for point of departure (POD) determination, as an alternative to the no observed adverse effect level (NOAEL). This approach was specifically chosen, as for the relevant neoplastic endpoints only lowest observed adverse effect level (LOAEL) values are available. The toxicological endpoint from the animal studies with the most conservative BMD lower confidence limit (BMDL) value was identified. Using the obtained BMDL value of 8 mg/kg body weight/day as POD, an acceptable daily intake (ADI) of 16 µg/kg body weight/day was obtained, which—despite being more conservative than previous approaches—is still clearly above the estimated daily exposure level to isoeugenol in the USA and in Europe. These results confirm a low risk of the estimated daily exposure levels of isoeugenol.

## 1. Introduction

Isoeugenol (4-hydroxy-3-methoxypropenylbenzene, see Figure 1) is part of a group of propenylbenzenes that occur naturally in numerous plants frequently employed as food constituents. Among others, isoeugenol can be found in basil, nutmeg and cloves. Further prominent members of this group are eugenol, safrole, estragole and methyleugenol. The primary detoxification pathway for isoeugenol involves the conjugation of its free phenolic group with sulfate or glucuronic acid, followed by the excretion of these metabolites in the urine [1].

Isoeugenol is produced industrially and used as a fragrance or flavoring agent in cosmetics, personal hygiene products, household cleaning agents and food. Furthermore, it is authorized as a feed additive as well as in veterinary medicines to anesthetize fish. Hence, the exposure of the general population to isoeugenol occurs via food and consumer products (perfumes, household cleaning agents, cigarette smoke) [2,3].

The exposure to isoeugenol via food was calculated by the Joint FAO/WHO Expert Committee on Food Additives (JECFA) in 2004 for Europe and the USA [4]. The calculation considered the intake via artificially added isoeugenol to food as flavoring agent and naturally occurring isoeugenol in plant food. According to JECFA, the intake of isoeugenol in the USA via naturally occurring food is about seven times higher than the intake due to artificially added isoeugenol as a flavoring substance. The JECFA estimated that the total isoeugenol levels via food intake were 117 µg per day and per capita in Europe and 43 µg per day and per capita in the USA. For a more accurate comparison, the total intake per kg was calculated using a body weight (b.w.) of 60 kg, resulting in 1.95 µg isoeugenol/kg b.w./day in Europe and 0.72 µg isoeugenol/kg b.w./day in the USA.

The exposure to isoeugenol via consumer products is given in a fragrance ingredient safety assessment for isoeugenol, which was performed by the Research Institute of Fragrance Materials (RIFM) [5] using a special calculation method from the fragrance industry. This method resulted in an exposure level of isoeugenol from consumer products of 0.4 µg/kg b.w./day. Since the value’s applicability to either Europe or the USA was not specified, it was assumed valid for both regions. Consequently, the value of 0.4 µg/kg b.w./day was added to the isoeugenol exposure from food as estimated by JECFA for Europe (1.95 µg/kg b.w./day) and the USA (0.72 µg/kg b.w./day). This results in a cumulative exposure of 2.35 µg isoeugenol/kg b.w./day in Europe and 1.12 µg isoeugenol/kg b.w./day in the USA. The exposure levels are summarized in Table 1.

In the EU, isoeugenol is listed in the union list of flavorings (Annex I of Regulation 1334/2008) [6] and can be used as flavoring substance without limitation. In the USA, isoeugenol has the so-called GRAS status, which means that it is generally recognized as safe as a food ingredient. Hence, in both regions, isoeugenol can be added to food without restriction.

However, there are some restrictions concerning the use of isoeugenol in consumer products in the EU due to its skin-sensitizing properties. For the use of isoeugenol in cosmetic products, it is required that isoeugenol is labelled as an ingredient when its concentration exceeds the limit that is stated in Annex II of the Cosmetic Regulation No. 1223/2009 [7]. Furthermore, it is included in the list of allergenic fragrances that should not be used in toys (Directive 2009/48 on the safety of toys) [8].

In veterinary medicine, isoeugenol is approved as a fish anesthetic and a respective maximum residue level (MRL) value for isoeugenol in fish was set in an MRL assessment report [9].

**Table 1 toxics-11-00991-t001:** Overview of published exposure estimations of isoeugenol.

	[µg/day]	[µg/kg b.w./day]
Exposure via Food (Artificially and Naturally)		
Europe (total intake in food) estimated by JECFA [4]	117	1.95 ^1^
USA (total intake in food) estimated by JECFA [4]	43	0.72 ^1^
- due to its addition as a flavoring agent (USA)	-	0.10 ^2^
- due to its natural occurrence in plants (USA)	-	0.62 ^2^
Exposure via Consumer Products		
Total systemic exposure (inhalation, oral, dermal, estimated by RIFM [5])	-	0.4
Total Exposure (Food and Consumer Products)		
Europe	-	2.35
USA	-	1.12

^1^ Calculation of the authors based on a body weight of 60 kg, as used by JECFA. EFSA uses a conversion factor of 70 kg [10]. ^2^ Calculated by the authors of this article from the total intake from food in the USA (0.72 µg/kg b.w./day) due to the JECFA statement that the intake of isoeugenol due to its natural occurrence in plants is about seven times higher than the intake due to artificially added isoeugenol as flavoring substance (see Table 2 in section 2 of the WHO food additives series no. 52 from JECFA [4]).

The WHO International Agency for Research on Cancer (IARC) monographs program [2,3] recently evaluated the hazard of isoeugenol to cause cancer. As a result of this evaluation, the IARC assigned isoeugenol to Group 2B of the IARC hazard classifications, meaning that isoeugenol must be considered possibly carcinogenic to humans. This classification is based on neoplastic findings in experimental animal studies (2-year mouse and rat gavage studies), which were performed by the National Toxicology Program (NTP) [11].

For risk assessment, an LOAEL (lowest observed adverse effect level) value of 75 mg/kg/ day can be derived from these studies due to observed neoplastic liver changes in the form of hepatocellular adenoma and carcinoma. However, nowadays a BMDL (lower confidence limit of the benchmark dose) value is preferred to be used instead of an LOAEL value as POD (point of departure) for risk characterization [12]. Hence, in this study, several relevant endpoints from the above-mentioned 2-year mouse and rat gavage studies were used to calculate a BMDL value, which can be used as POD for the risk characterization of isoeugenol.

## 2. Materials and Methods

The data used for BMD (benchmark dose) modeling were taken from the NTP Technical Report No. 551 “Toxicology and carcinogenesis studies of isoeugenol in F344/N rats and B6C3F1 mice” [11]. In detail, gavage studies of 3 months and 2 years were performed on rats and mice. As the test substance, an isomer mixture was used that consisted of 87% E-isoeugenol and 12% Z-isoeugenol.

The BMD modeling approach uses different mathematical models to calculate a regression curve for experimentally determined dose–response values of different study endpoints. For this approach, the United States Environmental Protection Agency (EPA) software BMDS (software version 3.3 [13], technical guidance [14]) offers different mathematical functions (also referred to as models) to calculate these regression curves. Continuous data (body weight) or dichotomous data (incidences) can be used for this approach.

In addition to the regression curve, a benchmark response factor (BMRF) must be defined. The BMRF value is the so-called extra risk. The extra risk is the minimal increase in an adverse effect in comparison to the control value that just results in a recognizable increase in the investigated endpoint compared to the control. Due to experiences with historical data, an extra risk of 10% is recommended by the EPA. The value on the x-axis where the sum of the control value and the extra risk (displayed on the y-axis) intersects with the calculated regression curve is the BMD value (dose value that just evokes a measurable response in the assay, e.g., animal study). To minimize the risk, the calculated BMD value is not used as POD, but rather, its lower confidence limit, the BMDL value.

Before the run of a new calculation, the user can change several settings, depending on the kind of data (continuous or dichotomous). The settings used for the calculations in this article are listed in Table 2.

**Table 2 toxics-11-00991-t002:** Settings applied with the BMDS software.

Dichotomous Data (Quantal Data)	
Analysis type ^1^	Frequentist
Models used	All available models
Restriction ^2^	The available models were used in their restricted and unrestricted form if both options were offered
Maximum degree for the multistage model	One less than the number of test groups (including control) but not more than 3
Risk type	Extra risk
BMRF	0.1 (10%)
Confidence level	0.95
Continuous Data	
Analysis type ^1^	Frequentist
Models used	All available models
Restriction ^2^	The available models were used in their restricted and unrestricted form if both options were offered
Maximum degree for the polynomial model	One less than the number of test groups (including the control group) but not more than 3
Risk type	Relative deviation
BMRF	0.1 (10%)
Confidence level	0.95
Adverse direction	Depending on the endpoint (the direction is “down” for body weight)
Distribution	Normal
Variance	Constant

^1^ Analysis type: For calculating the curves of the different models, two different statistical approaches can be used: a frequentist or a Bayesian approach. The Bayesian approach has more recently been implemented in the BMDS software. According to the current user guide for the EPA BMDS software (Version 3.3 from July 2022 [13]), the EPA currently does not offer technical guidance for Bayesian modeling. Hence, no reference values were offered for the Bayesian approach to see how good the available mathematical models fit the data (for the frequentist models, the quality of fit is given by the *p*-value and the Akaike Information Criterion (AIC) value). Therefore, the authors decided not to use the Bayesian approach. ^2^ Restriction: For some of the mathematical functions, a restricted and an unrestricted version is available. Restriction means that a special parameter of a function is restricted to pre-set values to avoid biologically implausible curves.

The results from the different models are classified into three different categories with the EPA’s BMDS software: unusable, questionable or viable. Only results from viable models were recommended by the software for use as a POD. The most important parameter for the classification of a model as questionable or viable is the goodness-of-fit *p*-value. A small *p*-value indicates a poor fit of the curve to the measured response values. The default *p*-value is *p* ≥ 0.10. A model with a *p*-value below 0.10 is not considered viable. If more than one viable model is available, the software recommends one of these models due to the following criteria:From the viable models with a “sufficiently narrow range”, those with the lowest AIC value will be selected [14].Otherwise, the model with the lowest BMDL value is recommended by the software.

If the software recommended a viable model, the BMD/BMDL values from this model were used for the respective endpoint until there was a scientific reason for another decision.

The BMDL value, the BMD value, the *p*-value, and the AIC value from the viable models are given in the results section. The raw results spreadsheet (Appendix A) from the BMDS software with the detailed data is available in the Appendix A.

## 3. Results

The modelled data included the observed neoplastic endpoints that resulted in the classification of isoeugenol as possibly carcinogenic by the IARC. In addition, BMD values were calculated for the observed decrease in body weight and non-neoplastic changes in the nose epithelium.

### 3.1. BMD Modeling of Neoplastic Liver Lesions

Statistically significantly increased neoplastic liver lesions were noted in the form of adenoma and carcinoma for the male mice of all dose groups (75, 150 and 300 mg/kg b.w./day) from the 2-year gavage studies published in the NTP Technical Report No. 551. In contrast, no increased neoplastic liver lesions were noted for the female mice and the male and female rats.

The incidences of adenomas and carcinomas (alone and combined) used for BMD modeling are given in Table 3. A clear dose–response relationship is not visible. The incidences at the low, the intermediate and the high dose level were between 66% and 74% for the endpoint of hepatocellular adenoma, between 36% and 38% for the endpoint of hepatocellular carcinoma and consistently at 86% for the endpoint of hepatocellular adenoma and carcinoma combined (Table 3).

All three endpoints given in Table 3 were used for BMD modeling. No viable model was received from the combination of hepatocellular adenoma and carcinoma (see Appendix A), one viable model was noted for the endpoint of hepatocellular adenoma (see Table 4) and nine viable models for the endpoint of hepatocellular carcinoma (see Table 5).

In the case of hepatocellular adenoma, only the unrestricted grade 2 multistage model (the two b-parameters are unrestricted; the number of b-parameter reflects the grade of the multistage model, see Appendix A) was judged as a viable model by the BMDS software. The calculated curve fits well with the data points. However, the good fitting of the unrestricted grade 2 multistage model with the data points is at the expense of biological plausibility, as the curve decreases from the intermediate dose level to the high dose level (see Figure 2). The use of only one restricted b-parameter by bounding the other b-parameters to zero (as carried out by the restricted grade 2 and grade 3 multistage models or the unrestricted grade 1 multistage model) results in a linear curve that increases from the control dose to the high dose level (see Figure 3) and a significantly increased BMDL value of 33 mg/kg b.w./day instead of 8 mg/kg b.w./day (see Table 4). However, the restricted models failed the fitting criteria, as their *p*-value was below 0.1.

Hence, both possibilities (using the unrestricted grade 2 multistage model or using models with only one unrestricted parameter) showed disadvantages (a biologically implausible curve or a *p*-value below the recommended value of the software). However, the unrestricted grade 2 multistage model was chosen due to the following reasons:It has the more conservative value.The fitting of the unrestricted curve near the BMD value is good (low values for the scaled residuals: −0.5 instead of 1.19 for the data point near the BMD value; see Appendix A) and the decreasing course of the curve near the high-dose group is not important for the determination of the BMD/BMDL values, which were taken between the control and the low-dose group.

Also, for the endpoint of hepatocellular carcinoma, the software recommended the unrestricted grade 2 multistage model with a BMDL value of 18 µg/kg b.w./day (Table 5). As for the endpoint of hepatocellular adenoma, the curve decreased from the intermediate to the high dose level (see Figure 4). However, the model was chosen due to its good fit between the control group and the low-dose group, which is the area of the curve from which the BMD and the BMDL values were determined.

For the endpoint of hepatocellular adenoma and carcinoma combined, an additional approach was used, skipping the high-dose group (300 mg/kg b.w./day). According to section 2.3.6 of the Technical Guidance of the BMDS software, it is permitted to omit the highest-dose group if none of the available models provide an adequate fit (no model available with a *p*-value ≥ 0.1) [14]. The omitting of the high-dose group can be justified by the fact that there is no dose–response relationship between the dose groups (same incidence of 86% for all dose groups; see Table 3) and that the BMDL value is near the low-dose group. For this approach, seven viable models were noted (Table 6). For example, the graph from the restricted gamma model is shown in Figure 5. As for the endpoint of hepatocellular adenoma, a BMDL value of 8 mg/kg b.w./day was found.

### 3.2. BMD Modeling of Other Neoplastic Lesions: Histiocytic Sarcomas, Thymoma and Mammary Gland Carcinoma

A histiocytic sarcoma is a rare neoplastic lesion in humans that is preferentially noted in the skin, soft tissue and gastrointestinal tract. It derives from cells of the monocyte/macrophage lineage [16].

Slightly increased incidences compared to the control group were noted for the endpoints of histiocytic sarcomas, thymoma and mammary gland carcinoma in the 2-year gavage studies performed for the NTP program. A statistical analysis revealed a positive dose-related trend for all three endpoints (*p* ≤ 0.05). Together with the observed neoplastic liver lesions, the observed statistically significant positive trends for these three neoplastic lesions led to the IARC classification of isoeugenol as possibly carcinogenic. As no NOAEL values were published for these three endpoints, the authors of this article also tried to set suitable NOAEL values to compare these NOAEL values with the obtained BMDL values.

For the endpoint of histiocytic sarcomas, one female mouse each was noted at the low and the intermediate dose level and four female mice with histiocytic sarcomas were noted at the high dose level of the 2-year female mice gavage study. The increased incidence at the high dose level (8%) was at the upper range of the historical incidence for histiocytic sarcomas (0 to 8%), whereas the incidences at the low and the intermediate dose level (2%) were clearly within the historical range (see Table 7). However, due to the statistically significant positive dose-related trend, an NOAEL value at the intermediate dose level of 150 mg/kg b.w./day was considered suitable.

In case of the endpoint of thymoma, two affected male rats were noted at the high dose level, whereas no observations were made in the control group and the low and intermediate dose groups, resulting in a weakly positive dose-related trend. The incidence at the high dose level (300 mg/kg b.w./day) of 4% was slightly above the historical incidence of 2% for male rats with a thymoma (see Table 7). As the incidence at the high dose level was above the historical incidence, the authors considered an NOAEL at the intermediate dose level (150 mg/kg b.w./day) as justified.

Two male rats with a carcinoma of the mammary glands were noted at the high dose level, whereas no observation was noted at the control group and the low and intermediate dose groups, resulting in a weakly positive dose-related trend (Table 7). Due to the increase from the intermediate to the high dose level, the authors set the NOAEL to the intermediate dose level. However, historical control data for an assessment of the observed incidence at the high dose level were not available.

BMD modeling using the EPA software revealed similar results for all three endpoints (histiocytic sarcomas, thymoma, mammary gland carcinoma). All tested models from all three endpoints were classified as viable and for all three endpoints the restricted grade 3 multistage model was recommended based on the lowest AIC value (see Table 8 and Figure 6, Figure 7 and Figure 8). However, all models were provided with the comment “BMD value above highest dose”, as for all three endpoints the BMD values were above the high dose level of 300 mg/kg b.w./day. As for the BMD values, the BMDL values were also above (307 or 311 mg/kg b.w./day for thymoma and mammary gland carcinoma) or near the highest dose level (239 mg/kg b.w./day for histiocytic sarcomas) (see Table 8).

Hence, for all three endpoints, the calculated BMDL value was above the NOAEL value of 150 mg/kg b.w./day (the suitable NOAEL according to the opinion of the authors of this article).

### 3.3. BMD Modeling of Non-Neoplastic Lesions of the Nose

Non-neoplastic lesions of the nose were noted for the rats and/or mice in the 3-month and 2-year gavage studies in the form of olfactory epithelial atrophy, olfactory epithelial respiratory metaplasia, degeneration of the olfactory epithelium, hyaline droplet accumulation in the olfactory epithelium and/or hyperplasia of the nasal glands. BMD modeling was performed for all observations with statistically significant changes and the obtained BMDL values are given in Table 9. The smallest BMDL value was found for the 3-month rat gavage study for an atrophied olfactory epithelium in female rats (5 mg/kg b.w./day). The obtained results from the BMD modeling of this endpoint are given in Table 10 and the related curve can be seen in Figure 9.

The obtained BMDL value of 5 mg/kg b.w./day is below the low dose level of 37.5 mg/kg b.w./day. For this endpoint (atrophied olfactory epithelium), the low dose level can be taken as the LOAEL, as the authors of this article believe that the observed incidences at the low dose level (30% for the male rats and 10% for the female rats) are a significant increase in comparison to the control group, where no rats with an atrophied epithelium were noted in the 3-month rat gavage study.

### 3.4. BMD Modeling of Body Weight

Slightly reduced body weights were noted for male rats and male mice at the end of the 3-month gavage studies using dose levels of 37.5, 75, 150, 300 and 600 mg/kg b.w./day. The effect was slightly more pronounced for the rats than for the mice (Table 11). Therefore, rat values were utilized for BMD modeling.

For the male rats, a slightly but continuously reduced body weight was noted from a dose level of 150 mg/kg b.w./day (336 ± 7 g) onwards up to the high dose level of 600 mg/kg b.w./day (307 ± 7 g) in comparison to a body weight of 352 ± 8 g, which was noted for the male rats in the control group. The body weights in the two low-dose groups were slightly below (325 ± 3 g at 37.5 mg/kg b.w./day) or in the range (334 ± 6 g at 75 mg/kg b.w./day) of the body weights noted at the intermediate dose level of 150 mg/kg b.w./day (336 ± 7 g).

Using all five non-zero dose levels for benchmark modeling, all models had a *p*-value below 0.1 and were classified as questionable. However, as a continuous and dose-related reduction in body weight was only noted from a dose level of 150 mg/kg b.w./day onwards (see Table 11), the two low dose levels (37.5 and 75 mg/kg b.w./day) were excluded from benchmark modeling. Running a BMDS analysis without the two low dose levels revealed several viable models and the software recommended the unrestricted power model (BMD = 438 mg/kg b.w./day; BMDL = 372 mg/kg b.w./day) due to the lowest AIC value (Table 12 and Figure 10).

When not excluding the two low dose levels, BMDL values of 130 mg/kg b.w./day (including the 75 mg/kg b.w./day dose level) and 401 mg/kg b.w./day (including the 37.5 and the 75 mg/kg b.w./day dose levels) were obtained using the model with the lowest BMDL value. However, as noted above, in this case all models were classified as questionable due to a *p*-value below 0.1.

An NOAEL value for the endpoint of body weight could not be found in the literature. The authors are of the opinion that the reductions in body weight noted at 300 and 600 mg/kg b.w./day (minus 7.4 and minus 12.8%) are of toxicological relevance. The observed decrease at 150 mg/kg b.w./day (minus 4.5%) is too small and the observed reduction at 37.5 mg/kg b.w./day (minus 7.8%) reveals no dose–response relationship. According to the authors of this article, an NOAEL value of 150 mg/kg b.w./day is plausible for the endpoint of body weight. Hence, the obtained BMDL value of 372 mg/kg b.w./day was above the NOAEL value.

## 4. Discussion

### 4.1. Summary of Calculated BMDL Values

The calculated BMDL values are summarized in Table 13. To demonstrate the difference between the NOAEL approach and the BMD approach, LOAEL and NOAEL values are shown additionally in Table 13 for the investigated endpoints.

The most conservative value was obtained for the endpoint of an atrophied olfactory epithelium (5 mg/kg b.w./day). Although several authors mentioned the nose findings, the only available toxicological assessment was from the Committee for Veterinary Medicinal Products (CVMP). The CVMP suggested that the effects on the nose were a result of local irritation due to contact with the test material rather than systemic exposure [9]. A local irritation in the nose is plausible for a volatile flavor compound such as isoeugenol, which might lead to co-exposure from inhalation. However, this might also happen in humans exposed to isoeugenol. To the authors’ knowledge, the nose lesions are currently not considered informative for risk assessment and were not used to calculate any toxicological reference value. Therefore, the BMDL value from this endpoint was not considered, and the second-most conservative value was utilized. This value was obtained from the endpoint of hepatocellular adenoma (8 mg/kg b.w./day) and also from the endpoint of hepatocellular adenoma and carcinoma combined.

Hence, a BMDL value of 8 mg/kg b.w./day was employed for ADI calculations. This BMDL value of 8 mg is clearly below the POD value of 75 mg/kg b.w./day that was used by the CVMP in a Maximum Residue Assessment Report of isoeugenol in fish in order to set an ADI for the establishment of an maximum residue level (MRL) for fish that were treated with isoeugenol (see section 4.4, ‘”Re-calculation of the MRL value for fin fish”) [9]. The POD of 75 mg/kg b.w./day set by the CVMP corresponds to an LOAEL value of 75 mg/kg b.w./day, which was obtained from the observed neoplastic liver lesions of the 2-year NTP gavage study, the same endpoint as the calculated BMDL value.

### 4.2. Rationale for the Approach for Risk Assessment of Isoeugenol

As mentioned above, the IARC has recently classified isoeugenol as possibly carcinogenic due to the observed neoplastic changes that were noted in 2-year gavage studies in mouse and rats. For the risk assessment of carcinogenic substances, their mode of action is essential, as they can cause their carcinogenic effect via a genotoxic or a non-genotoxic mechanism. Hence, according to their presumed mode of action, carcinogenic substances are categorized as genotoxic or non-genotoxic.

Genotoxicity is defined as the capability to cause mutations [17,18] and/or cellular DNA damage (chromosomal aberrations) in the form of structural chromosomal aberrations (clastogenicity) or numerical chromosome aberrations (aneugenicity) [19,20,21]. A non-genotoxic carcinogen can theoretically be every substance that induces a proliferative process (e.g., mitogens), because the possibility that a mutation will occur spontaneously increases with the frequency of cell divisions [22,23].

Genotoxicity itself can be sub-divided into direct and indirect genotoxicity [24]. Direct genotoxicity is caused by genotoxic substances that directly interact with the DNA. This interaction can occur by DNA adduct formation, either by the original substance or after metabolic activation of the substance [25]. Indirect mechanisms of genotoxicity are caused by the interaction of a genotoxic substance with non-DNA targets, such as topoisomerases or spindle proteins [26].

Based on their mode of action, carcinogenic substances can be further divided into substances for which a threshold value can be identified or not. A threshold value means that a dose concentration exists below which toxicity does not occur, whereas non-threshold means that there is no dose below which cancer initiation does not take place because the mode of action may involve a single direct reaction (single hit at a single target) [27]. Direct genotoxic carcinogens are traditionally considered as non-thresholded carcinogens (even if they need a metabolic activation), whereas a threshold is conceivable for indirectly acting genotoxic carcinogens, as shown for substances that interact with spindle proteins (see below). Non-genotoxic carcinogens are generally considered to have a threshold value. However, it is now increasingly accepted that thresholds for direct carcinogens could be detected at low concentrations. This could be due to the activity of repair enzymes or the need for metabolic activation. Nevertheless, from a regulatory point of view, linear (non-threshold) models are still assumed to apply for direct genotoxic substances [27,28].

The differentiation between thresholded and non-thresholded substances is important for risk assessment. If an exposure level is expected that will be safe (threshold substance), an ADI value (acceptable daily intake) can be calculated. The ADI value represents an estimate of the amount of a substance in food or drinking water that can be consumed daily over a lifetime without presenting an appreciable risk to health [29].

As noted above, no threshold values were expected for direct genotoxic carcinogens (possible single hit at a single DNA base) and the regulations for these substances may be based on the as low as reasonably achievable (ALARA) principle. To estimate the risk from such a non-thresholded genotoxic carcinogen, a margin of exposure (MOE) value may be calculated that is based on the exposure level of the carcinogen. The MOE value is the ratio between the POD and the exposure level. For genotoxic substances, a MOE value above 10,000 is considered to be of low concern for public health if it is based on a BMDL as POD [30,31,32].

Hence, to establish a scientifically valid risk assessment for isoeugenol, it is important to know if it acts via a genotoxic or a non-genotoxic mechanism. In the case of a genotoxic mechanism, it is furthermore important if the mechanism has a threshold or not. To better evaluate how a carcinogen causes cancer, the IARC evaluates ten key characteristics that are related to multiple mechanisms that can cause cancer [33]. To decide if an acceptable daily intake (ADI) value or an MOE value has to be used for risk assessment, the key characteristic “is genotoxic” is of particular importance. To make a decision about genotoxicity, the IARC has considered the following published studies that investigated the ability of isoeugenol to cause mutations and/or chromosomal aberrations.

Mutagenicity studies to test the mutagenic potential of isoeugenol were performed in bacteria using different AMES tests (*S. typhimurium* strains TA98, TA100, TA1535 and TA1537 and *Escherichia coli* strain WP2 *uvrA* with and without metabolic activation) [34] and in vivo using liver tissue from transgenic *gpt* delta mice (B6C3F1 *gpt* delta mice) [35]. Substances that are positive in these mutagenicity studies could be considered as a direct genotoxic substance without a threshold. However, the results for isoeugenol in these mutagenicity studies were all negative.

In contrast, studies to test if isoeugenol can cause mutations and/or chromosomal aberrations were ambiguous. An in vitro study for chromosomal aberrations in the form of sister chromatid exchange (SCE) in cultured Chinese hamster ovary cells (CHO K-1) was negative [36]. Another in vitro study revealed no induction of unscheduled DNA synthesis in isolated hepatocytes from male Fischer 344 rats and female B6C3F1 mice [37]. An in vivo mutagenicity and recombination study in *Drosophila melanogaster* (wing spot somatic and recombination test; SMART) was also negative [38]. Finally, an in vivo study using isoeugenol-treated turkey embryos revealed no DNA strand breaks (COMET assay) or DNA adduct formation in the fetal liver of the turkey embryos [39]. In contrast, in two different in vitro studies using freshly isolated human lymphocytes, isoeugenol was noted to induce chromosomal aberrations in the form of sister chromatid exchange (SCE) [40,41]. Furthermore, the ability of isoeugenol to induce DNA damage was noted in a *Bacillus subtitlis* DNA-repair test using the repair-competent strain H17^+^Rec^+^ and the repair-deficient strain M45 Rec^-^ (published together with the above-mentioned AMES test in *Escherichia coli*) [34]. Finally, a mouse peripheral blood micronucleus test was performed using the male and female mice (B6C3F1) that were used in the 3-month gavage study, which was performed during the NTP program (see Section 2). In this study, statistically significantly increased frequencies of micronucleated erythrocytes were noted in peripheral blood samples from the high-dosed females (600 mg/kg b.w./day) together with a positive trend test. In contrast, no statistically significant changes were noted in the blood samples from the male mice (see Table E3 in NTP Technical Report No. 551) [11].

As discussed above, a non-threshold mechanism is conceivable for indirectly acting genotoxic substances and is generally accepted for the chromosomal aberrations in the form of micronuclei (aneugenicity) [21,26,31,42]. In detail, a micronucleus can consist of chromosome fragments (clastogenicity) or a whole chromosome. The segregation of a whole chromosome into a micronucleus leads to an abnormal number of chromosomes in the cell. Such an aneuploidy that is caused by chemicals is called aneugenicity. Aneugenicity that is due to micronucleus formation is caused by inhibition of the mitotic/meiotic spindle. According to the International Conference on Harmonisation (ICH) guideline S2(R1) for the testing of genotoxicity, the inhibition of the spindle by a genotoxic substance is considered to be a threshold-dependent process [43].

According to the available data, the IARC came to the conclusion that the mechanistic evidence considered for the key characteristics of carcinogens (including genotoxicity) is sparse or the available results are negative in experimental systems.

In summary, the key question is whether isoeugenol evokes its possible carcinogenic activity via a genotoxic or a non-genotoxic mechanism. As discussed above, indications for a genotoxic mechanism in the form of chromosomal aberrations were noted in the form of SCE, a positive DNA-repair test in *B. subtilis*, and increased numbers of erythrocytes with a micronucleus in the peripheral blood of isoeugenol-treated female mice. The SCE study [40,41] and the DNA-repair test [34] in *B. subtilis* are nearly 40 years old and are not considered anymore in the current guidelines or were never part of an official Organization for Economic Co-operation and Development (OECD) test guideline (DNA-repair test in *B. subtilis*) for genotoxicity testing. The possible clastogenic property of isoeugenol, which was suggested by these two old studies, is disproved by the negative results from more-recently performed studies. In detail, as discussed above, no chromosomal aberrations were noted in Chinese hamster cells [11], no synthesis of unscheduled DNA was noted in rat or mice hepatocytes [37] and no DNA strand breaks were noted in the COMET assay [39].

What still needs to be clarified are the results of the positive in vivo micronuclei study. However, this clarification was already provided by the CVMP in an MRL assessment report of isoeugenol in fin fish in 2010 [9]. Faced with the unclear status of genotoxicity in a previous CVMP evaluation, the CVMP requested further genotoxic studies from the applicant in the form of an in vivo micronucleus test in male and female mice and an in vivo UDS (unscheduled DNA synthesis) test in male and female rats. Both tests were performed in vivo and revealed no genotoxic potential, thereby using higher isoeugenol concentrations, as in the 3-month gavage study, which showed the formation of micronuclei (see above). Hence, the CVMP concluded that the weight of evidence is sufficient to conclude that isoeugenol is not genotoxic. As these newly performed genotoxic studies are unpublished regulatory studies, they were not considered in the more current IARC assessment of isoeugenol. This might be the reason why the conclusion concerning the genotoxicity of isoeugenol differs between the IARC (mechanistic information about how isoeugenol causes cancer is sparse) and the CVMP (isoeugenol is not genotoxic).

However, considering all studies, the authors of this publication agree with the CVMP classification that isoeugenol is not genotoxic. Hence, the observed possible carcinogenic activity of isoeugenol is most likely due to a non-genotoxic mode of action and a threshold-based mechanism can be assumed. Accordingly, the calculation of an ADI is the correct approach for the risk assessment of isoeugenol.

### 4.3. Calculation of ADI Value

The CVMP established an ADI value in the MRL assessment report for fish that were anaesthetized with isoeugenol [9]. As POD for this ADI value, an LOAEL for isoeugenol of 75 mg/kg b.w./day was selected for the endpoint of neoplastic liver lesions, as noted above.

For the determination of the ADI by the CVMP, an uncertainty factor of 1000 was used, resulting in an ADI of 75 µg/kg b.w./day. The uncertainty factor of 1000 consists of an uncertainty factor of 100 for inter/intraspecies extrapolation, an uncertainty factor of 2 for the use of an LOAEL instead of an NOAEL and an uncertainty factor of 5 to account for the potential seriousness and irreversibility of the effect combined with the deficiencies of the 2-year gavage study [9].

However, as in this article a BMDL value was applied instead of an LOAEL value, the uncertainty factor of 2 could be skipped. This results in an overall uncertainty factor of 500 instead of 1000, as was used by the CVMP. This calculation resulted in an ADI of 16 µg/kg b.w./day (Table 14). This ADI value is clearly above the exposure levels from food in Europe and the USA. Even when considering the additional exposure from consumer products, the value of 16 µg/kg b.w./day is still above the estimated exposure levels in Europe (2.35 µg/kg b.w./day) and the USA (1.12 µg/kg b.w./day).

### 4.4. Re-Calculation of the MRL Value for Fin Fish

As noted above, the CVMP established an MRL value of 6000 µg/kg for the use of isoeugenol-treated fish in Europe. This value is based on an estimated daily per capita fish intake of 0.3 kg, an ADI of 75 µg/kg b.w./day and an average human body weight of 60 kg. Furthermore, the CVMP calculation assumes that the amount of isoeugenol intake from isoeugenol-treated fish should not exceed more than 40% of the acceptable daily per capita intake of isoeugenol.

Using an ADI of 75 µg/kg b.w./day and a human body weight of 60 kg, the total acceptable daily intake is 4500 µg per capita and day. Forty percent of 4500 µg is 1800 µg. Hence, the isoeugenol intake via fish should not exceed 1800 µg, resulting in an MRL value of 6000 µg/kg fish (1800 µg divided by 0.3 kg fish).

Using an ADI of 16 µg/kg b.w./day and a human body weight of 60 kg, the total acceptable daily intake of isoeugenol is only 960 µg per capita and day instead of 4500 µg per capita and day and forty percent of 960 µg is 384 µg per capita and day instead of 1800 µg per capita and day. Hence, the isoeugenol intake from isoeugenol-treated fish should not exceed 384 µg per capita and day. This results in an MRL value of 1280 µg isoeugenol/kg fish instead of 6000 µg isoeugenol/kg fish. Hence, the MRL appears not conservative enough using this refined toxicological approach and the authors suggest that the EU should re-assess the isoeugenol MRL with priority.

### 4.5. Relevance of the Observed Neoplastic Liver Lesions for Humans

In their rationale for classifying isoeugenol as possibly carcinogenic, the IARC refers to the observed neoplastic lesions in the 2-year gavage studies performed with mice and rats. As discussed above, these neoplastic lesions are the observed statistically significantly increased hepatocellular adenoma and carcinoma in male mice as well as the statistically significantly increased positive trends in mammary gland carcinoma, benign or malignant thymoma in male rats and histiocytic sarcomas (multiple sites) in female mice [2,3].

Prior research has left uncertainty about the implications of these findings for human beings. The CVMP classified the findings in their MRL assessment report as equivocal and their relevance for the human consumer as unclear. Nevertheless, they used the observed statistically significantly increased incidence of neoplastic liver lesions at the low dose level (75 mg/kg b.w./day) as POD for calculating an ADI [9].

In a safety assessment report of isoeugenol, the Flavor and Extract Manufacturers Association of the United States (FEMA) expert panel classified the mice liver tumors as not relevant to a human risk, as the male mice from the mouse strain used are known to have a high rate of spontaneous neoplastic liver lesions. Furthermore, the observed histiocytic sarcomas were classified as a high-dose phenomenon that only occurred against the background of other observations of systemic toxicity [44]. However, the FEMA panel did not refer to the observed neoplastic lesions in the form of mammary gland carcinomas and thymoma in rats. Hence, the authors of this article are of the opinion that the overall increase in different neoplastic lesions, evidenced by a statistically significant positive trend, supports the CVMP classification for these neoplastic lesions as ambiguous and not just as irrelevant for the human consumer.

## 5. Conclusions

The BMD-based approach revealed a POD of 8 mg/kg b.w./day. This POD was significantly below the LOAEL value of 75 mg/kg b.w./day, which was previously used as POD by the CVMP for the calculation of the ADI value. The use of the new BMD-based POD led to a significant reduction in the ADI value. The resulting new ADI value is still clearly above the estimated daily per capita intake of isoeugenol when considering the intake of isoeugenol via food alone or the total exposure to isoeugenol via food and consumer products.

## Figures and Tables

**Figure 1 toxics-11-00991-f001:**
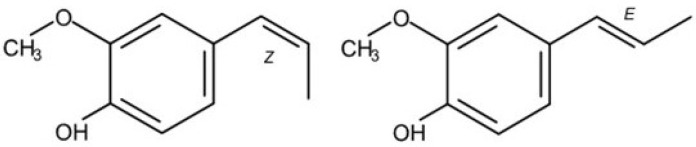
Chemical structure of isoeugenol (Z-(cis)-isomer, E-(trans)-isomer) [2].

**Figure 2 toxics-11-00991-f002:**
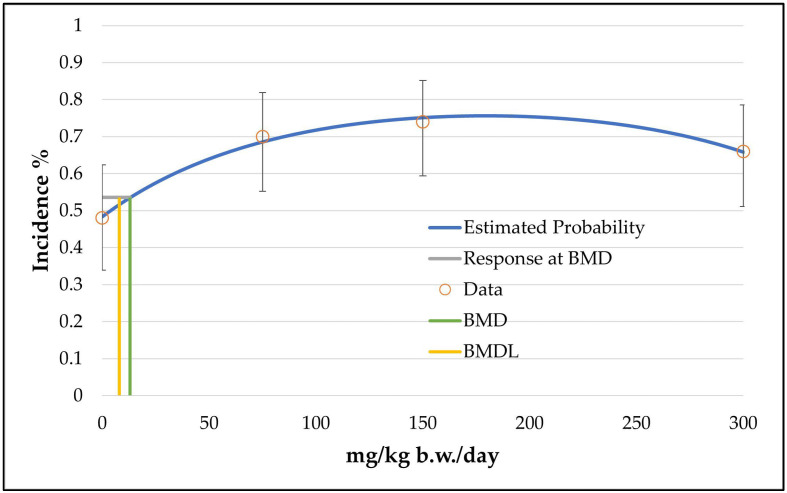
BMD model of the endpoint of hepatocellular adenoma in mice from the 2-year NTP gavage study. The graph shows the unrestricted grade 2 multistage model that was chosen for the determination of the BMD/BMDL value.

**Figure 3 toxics-11-00991-f003:**
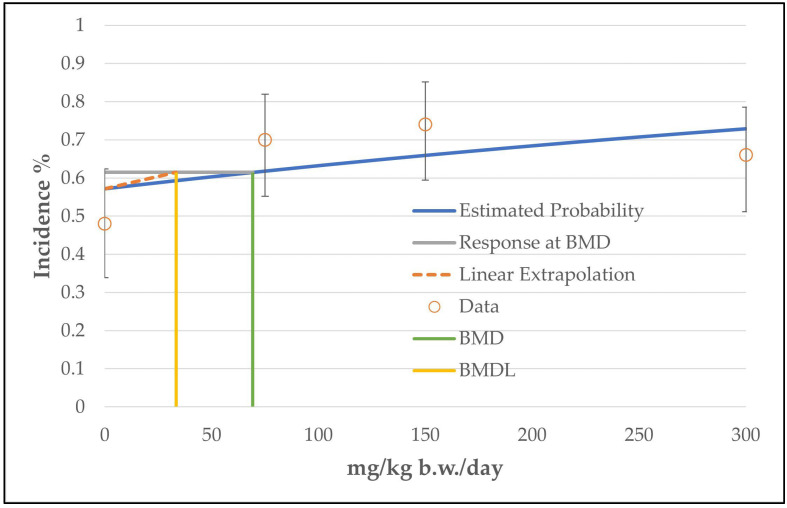
BMD model of the endpoint of hepatocellular adenoma in mice from the 2-year NTP gavage study. The graph shows the restricted grade 2 multistage model that was not chosen for the determination of the BMD/BMDL value, as the *p*-value is below 0.1. Note: The dashed line (linear extrapolation) is the so-called “cancer slope factor”. The line is a linear extrapolation from the POD to the origin, corrected for background (value of the control group). The slope of the line is the ratio between the BMR factor and the BMDL value and is given in multistage models when there is an indication that the dose–response curve has a linear component below the POD. The slope of the line describes an estimated risk per increment of dose [15]. The line is given by default and does not influence the BMDL value.

**Figure 4 toxics-11-00991-f004:**
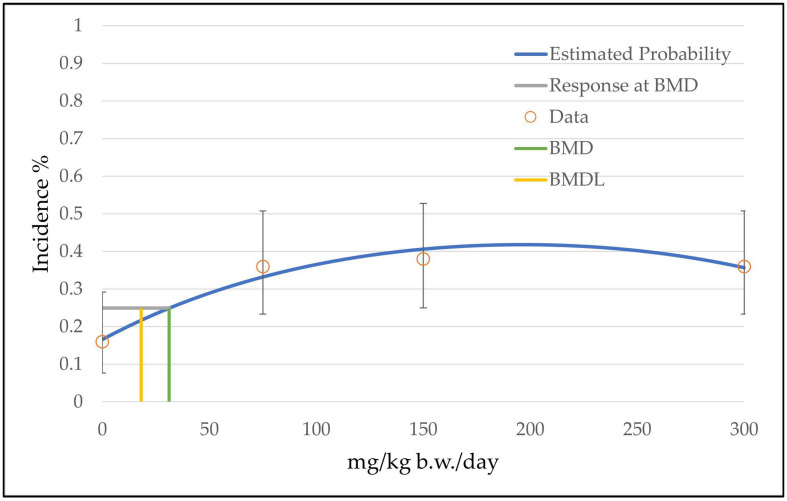
BMD model of the endpoint of hepatocellular carcinoma in mice from the 2-year NTP gavage study. The graph shows the unrestricted grade 2 multistage model that was chosen for the determination of the BMD/BMDL value.

**Figure 5 toxics-11-00991-f005:**
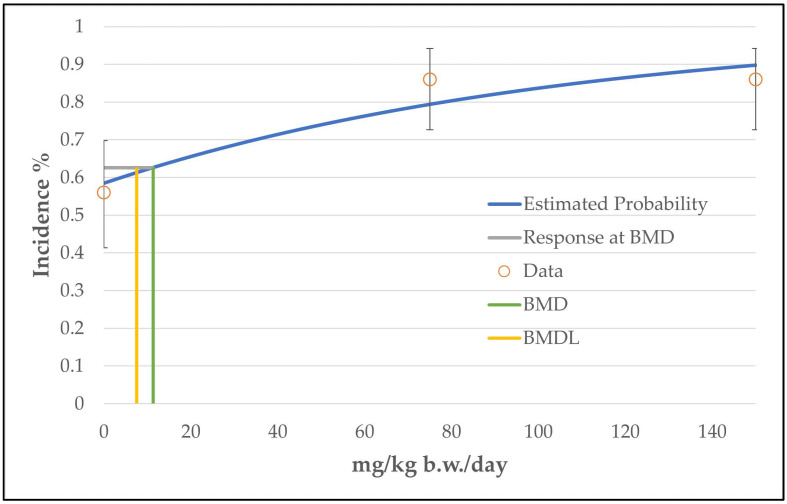
BMD model of the endpoint of hepatocellular adenoma and carcinoma combined in mice from the 2-year NTP gavage study. The graph shows the restricted gamma model that was chosen from the altogether 5 recommended models for the determination of the BMD/BMDL value (see Table 6).

**Figure 6 toxics-11-00991-f006:**
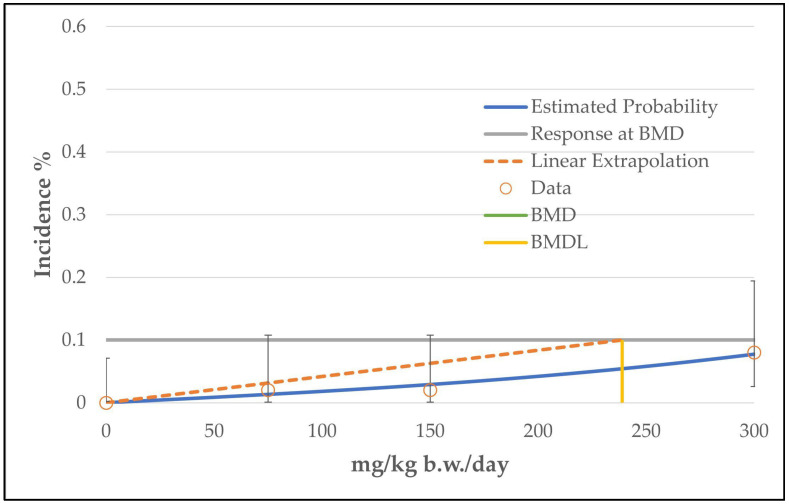
BMD model of the endpoint of histiocytic sarcomas in mice from the 2-year NTP gavage study. The graph shows the restricted grade 3 multistage model that was chosen for the determination of the BMD/BMDL value. The BMD value is above the highest dose level of 300 mg/kg b.w./day. For the meaning of the dashed line, see Figure 2.

**Figure 7 toxics-11-00991-f007:**
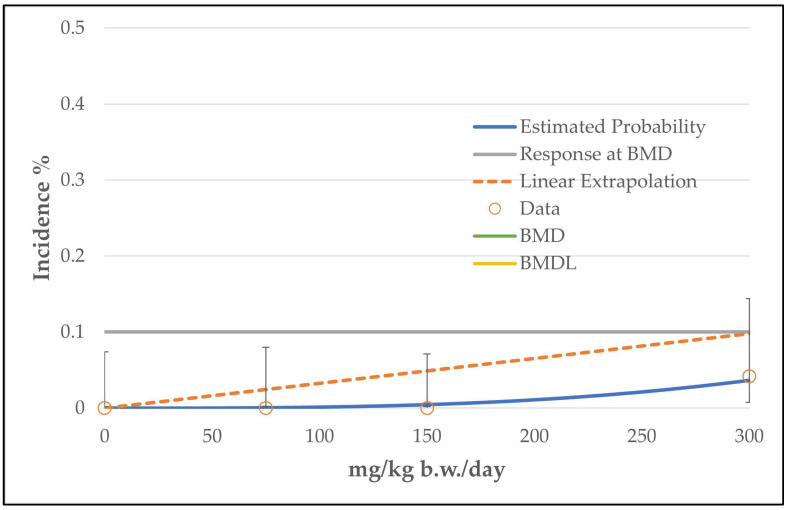
BMD model of the endpoint of thymoma in rats from the 2-year NTP gavage study. The graph shows the restricted grade 3 multistage model that was chosen for the determination of the BMD/BMDL value. The BMD and the BMDL value are above the highest dose level of 300 mg/kg b.w./day. For the meaning of the dashed line, see Figure 2.

**Figure 8 toxics-11-00991-f008:**
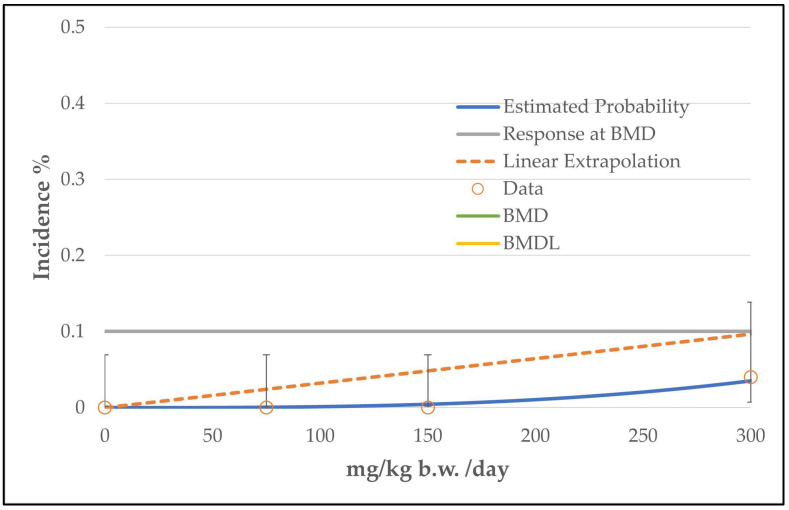
BMD model of the endpoint of mammary gland carcinoma in rats from the 2-year NTP gavage study. The graph shows the restricted grade 3 multistage model that was chosen for the determination of the BMD/BMDL value. The BMD and the BMDL value are above the highest dose level (300 mg/kg b.w./day). For the meaning of the dashed line, see Figure 2. Note to Figure 8: Figure 8 is nearly identical to Figure 7, as the incidences of thymoma and mammary gland carcinoma are identical (4% at the high dose level and 0% for the other groups; see Table 7 and the respective data sheets in the Appendix A).

**Figure 9 toxics-11-00991-f009:**
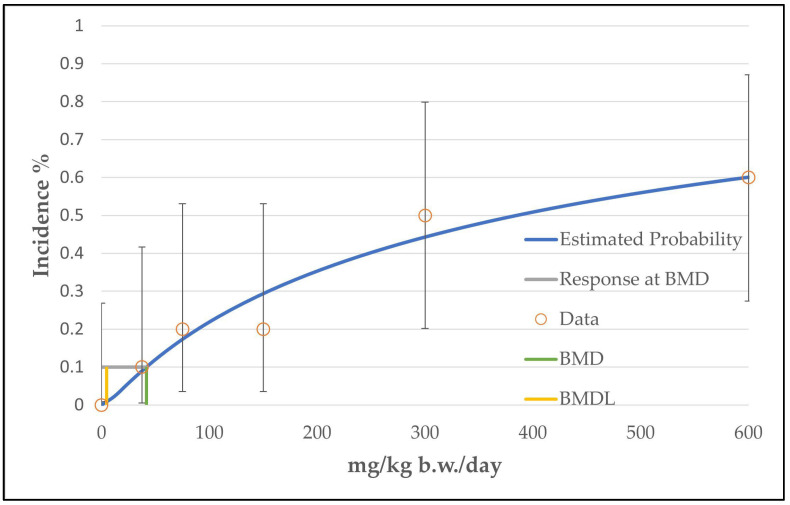
BMD model results of the endpoint of atrophied olfactory epithelium in male rats from the 3-month NTP gavage study. The graph shows the unrestricted log-probit model that was chosen for the determination of the BMD/BMDL value.

**Figure 10 toxics-11-00991-f010:**
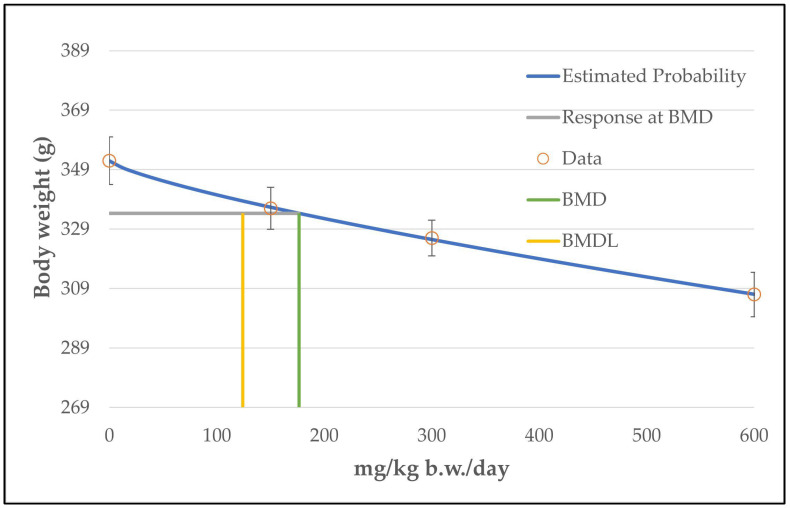
BMD model results of the endpoint of body weight in male rats from the 3-month NTP gavage study. The graph shows the unrestricted power model that was chosen for the determination of the BMD/BMDL value.

**Table 3 toxics-11-00991-t003:** Neoplastic liver lesions noted in male mice from a 2-year gavage study performed for the National Toxicology Program (NTP).

Neoplastic Liver Lesions Affected Animals/Total (Overall Rate in %) ^1^				
	Control	75 mg/kg b.w./day	150 mg/kg b.w./day	300 mg/kg b.w./day
Male mice				
Hepatocellular adenoma (including multiple) ^2^	24/50 (48%)	35/50 * (70%)	37/50 ** (74%)	33/50 ** (66%)
	*p* = 0.012 ^3^			
	(48 to 52%) ^4^			
Hepatocellular carcinoma (including multiple) ^2^	8/50 (16%)	18/50 * (36%)	19/50 * (38%)	18/50 * (36%)
	*p* = 0.027 ^3^			
	(16 to 28%) ^4^			
Hepatocellular adenoma and/or carcinoma ^5^	28/50 (56%)	43/50 ** (86%)	43/50 ** (86%)	43/50 ** (86%)
	*p* < 0.001 ^2^			
	(56 to 66%) ^4^			

*/**: Statistically significantly different from the control (* *p* ≤ 0.05; ** *p* ≤ 0.01) according to pairwise comparison between the control group and the respective dose group. For details of the statistical test, see NTP report 551 [11]. ^1^ The number of affected animals as well as the overall rate in percent are taken from Table 16 of NTP report 551 [11]. ^2^ The number of observed animals with liver lesions includes the animals with only one adenoma (or carcinoma) in the liver and the animals with multiple adenoma (or carcinoma) in the liver (for example, in the control group, 14 animals were noted with one adenoma in the liver and 10 animals with multiple adenoma in the liver, resulting in a total amount of 24 animals). ^3^ Result for the test of a dose-related trend. ^4^ Range of historical incidence for the vehicle control (Table 16 of NTP report 551 [11]). ^5^ Number of animals with an adenoma or a carcinoma alone and animals with both lesions combined.

**Table 4 toxics-11-00991-t004:** BMD modeling results for the endpoint of hepatocellular adenoma (including multiple).

Model	Goodness of Fit		BMD	BMDL	Rationale for Model Selection by BMDS Software
	*p*-Value	AIC			
Multistage Degree 2 (unrestricted)	0.77	257.82	13	8	The multistage degree 2 model was the only model with an adequate fit (*p* ≥ 0.1)
Multistage Degree 1 to 3 (restricted) Multistage Degree 1 (unrestricted)	0.06	261.56	69	33	Models classified as questionable, due to a *p*-value < 0.1

AIC = Akaike Information Criterion; BMD = benchmark dose, BMDL = lower confidence limit of the benchmark dose, BMDS = Benchmark Dose Software.

**Table 5 toxics-11-00991-t005:** Results of BMD modeling for the endpoint of hepatocellular carcinoma (including multiple).

Model	Goodness of Fit		BMD	BMDL	Rationale for Model Selection
	*p*-Value	AIC			
Log-Logistic R	0.15	248.80	88	43	From all models that provided an adequate fit (*p* ≥ 0.1), the unrestricted multistage degree 2 model was selected based on the lowest BMDL (unrounded values).
Gamma R,	0.13	249.22	108	58	
Multistage Degree 3 R,					
Multistage Degree 2 R,					
Multistage Degree 1 R,					
Weibull R,					
Quantal Linear UnR,					
Multistage Degree 1 UnR					
Multistage Degree 2 UnR	0.56	247.39	31	18	

R = restricted; UnR = unrestricted; AIC = Akaike Information Criterion; BMD = benchmark dose, BMDL = lower confidence limit of the benchmark dose.

**Table 6 toxics-11-00991-t006:** Results of BMD modeling for the endpoint of hepatocellular adenoma and carcinoma combined (the high-dose group of 300 mg/kg was skipped).

Model	Goodness of Fit		BMD	BMDL	Rationale for Model Selection
	*p*-Value	AIC			
Gamma R,	0.14	155.88	11	8	From all models that provided an adequate fit (*p* ≥ 0.1), the five models in bold were selected based on the lowest BMDL (unrounded values).
Multistage Degree 1 R,					
Weibull R,					
Multistage Degree 1 UnR,					
Quantal Linear					
Log-Probit R	0.12	155.94	20	13	
Multistage Degree 2 R ^1^	0.14	155.88	11	8	

R = restricted; UnR = unrestricted; AIC = Akaike Information Criterion; BMD = benchmark dose, BMDL = lower confidence limit of the benchmark dose. ^1^ For an unclear reason, the restricted multistage degree 2 model was not recommended by the software, though the values were the same as for the recommended models.

**Table 7 toxics-11-00991-t007:** Histiocytic sarcomas, thymoma and mammary gland carcinoma with a statistically significant positive dose-related trend noted in the 2-year gavage studies.

Histiocytic Sarcomas, Thymoma and Mammary Gland Carcinoma. Affected Animals/Total (Overall Rate in%) ^1^				
	0	75	150	300
Female mice				
Histiocytic sarcomas (multiple sites)	0/49	1/50 (2%)	1/50 (2%)	4/50 (8%)
	*p* = 0.015 ^2^			
	(0 to 8%) ^3^			
Male rats				
Thymoma (benign or malignant)	0/47	0/43	0/49	2/48 (4%)
	*p* = 0.047 ^2^			
	(0 to 2%) ^3^			
Carcinoma of the mammary gland	0/50	0/50	0/50	2/50 (4%)
	*p* = 0.042 ^2^			
	(not available) ^3^			

^1^ The number of affected animals, the overall rate and the range of the historical incidence are taken from Table 17 (histiocytic sarcoma), Table 8 (thymoma) and Table A2 (mammary gland carcinoma) of the NTP report 551 [11]. ^2^ Result for the test of a dose-related trend. ^3^ Range of historical incidence for the vehicle control.

**Table 8 toxics-11-00991-t008:** Results of BMD modeling for the endpoints of histiocytic sarcomas, thymoma and mammary gland carcinoma.

Model	Goodness of Fit		BMD	BMDL	Rationale for Model Selection by BMDS Software
	*p*-Value	AIC			
Histiocytic sarcomas					All models were recommended as viable for all 3 endpoints; the selected models were chosen due to the lowest AIC value.
Multistage Degree 3, R	0.96	49.80	348	239	
Range of other viable models	0.61 to 1.00	49.86 to 53.48	318 to 448	198 to 254	
Thymoma					
Multistage Degree 3, R	0.99	17.17	425	307	
Range of other viable models	0.68 to 1.00	17.73 to 20.87	308 to 1300	294 to 490	
Mammary gland carcinoma					
Multistage degree 3, R	0.99	17.33	430	311	
Range of other viable models	0.68 to 1.00	17.90 to 22.79	310 to 1367	228 to 516	

R = restricted; AIC = Akaike Information Criterion; BMD = benchmark dose, BMDL = lower confidence limit of the benchmark dose, BMDS = Benchmark Dose Software.

**Table 9 toxics-11-00991-t009:** Non-neoplastic lesions of the nose with statistically significant changes from 3-month and 2-year gavage studies in mice and rats.

		Affected Animals/Total Examined Animals (mg/kg b.w./day)						
	T ^1^	0	37.5	75	150	300	600	BMDL
Atrophy of the olfactory epithelium								
Male rats	4	0/10	3/10	3/10	4/10 *	4/10 *	5/10 *	9
Female rats	4	0/10	1/10	2/10	2/10	5/10 *	6/10 **	5
Male rats	9	1/50	-	5/48	9/49 **	13/49 **	-	44
Female rats	9	0/50	-	0/49	0/49	4/49 *	-	265
Male mice	12	0/10	0/10	0/10	0/10	0/10	10/10 **	297
Female mice	12	0/10	0/10	0/10	0/10	0/10	10/10 **	297
Male mice	18	5/50	-	13/50 *	36/50 **	41/50 **	-	- ^2^
Female mice	18	3/48	-	8/50	36/50 **	43/50 **	-	- ^2^
Metaplasia of the respiratory olfactory epithelium								
Male rats	9	4/50	-	6/48	10/49 **	15/49 **	-	18
Female rats	9	5/50	-	5/49	9/49	12/49 *	-	38
Male mice	18	4/50	-	31/50 **	47/50 **	49/50 **	-	15
Female mice	18	6/48	-	37/50 **	49/50 **	50/50 **	-	12
Degeneration of the olfactory epithelium								
Male rats	9	1/50	-	0/48	2/49	6/49 *	-	216
Male mice	18	1/50	-	1/50	7/50 *	6/50 *	-	135
Accumulation of hyaline droplets in the olfactory epithelium								
Male mice	18	0/50	-	6/50 *	26/50 **	19/50 **	-	49
Female mice	18	0/48	-	4/50	18/50 **	12/50 **	-	54
Hyperplasia of the nasal glands								
Male mice	18	3/50	-	34/50 **	49/50 **	48/50 **	-	- ^2^
Female mice	18	6/48	-	38/50 **	49/50 **	49/50 **	-	8

*/**: Statistically significantly different from the control (*: *p* ≤ 0.05; **: *p* ≤ 0.01) according to pairwise comparison between the control group and the respective dose group. For details of the statistical test, see NTP report 551 [11]. ^1^ Table number from NTP report 551, from which the number of animals and the statistical results are taken. ^2^ No viable model available.

**Table 10 toxics-11-00991-t010:** BMD modeling results for atrophied olfactory epithelium from female rats from a 3-month gavage study.

Model	Goodness of Fit		BMD	BMDL	Rationale for Model Selection by BMDS Software
	*p*-Value	AIC			
Dichotomous Hill R	0.96	58.45	41	14	From all models that provided an adequate fit (*p* ≥ 0.1), the unrestricted log-probit model was selected based on the lowest BMDL.
Log-Logistic R	0.99	56.45	42	24	
Log-Probit R	0.71	60.48	107	65	
Logistic UnR	0.56	61.81	147	103	
Gamma R,	0.95	56.91	55	37	
Multistage Degree 3 R,					
Multistage Degree 2 R,					
Multistage Degree 1 R,					
Weibull R,					
Quantal Linear UnR,					
Multistage Degree 1 UnR					
Log-Probit UnR	0.96	58.49	42	5	
Multistage Degree 3 UnR	0.90	60.43	44	18	
Multistage Degree 2 UnR	0.97	58.44	43	24	
Probit UnR	0.59	61.55	138	99	

R = restricted; UnR = unrestricted; AIC = Akaike Information Criterion; BMD = benchmark dose, BMDL = lower confidence limit of the benchmark dose, BMDS = Benchmark Dose Software.

**Table 11 toxics-11-00991-t011:** Mean body weights from male mice and rats from 3-month gavage studies.

Mean Body Weights at Study Termination (g)						
Body Weight	0	37.5	75	150	300	600
Male rats ^1^	352 ± 8	325 ± 3 *	334 ± 6 *	336 ± 7 *	326 ± 6 **	307 ± 7 **
		−7.7% ^2^	−5.1% ^2^	−4.5% ^2^	−7.4% ^2^	−12.8% ^2^
Male mice ^3^	37.7 ± 0.9	36.0 ± 1.0	36.1 ± 1.6	35.7 ± 1.1	37.1 ± 1.2	33.1 ± 1.0 *

*/**: Statistically significantly different from the control (*: *p* ≤ 0.05; **: *p* ≤ 0.01) according to pairwise comparison between the control group and the respective dose group. For details of the statistical test, see NTP report 551 [11]. ^1^ Mean body weights and statistical results were taken from Table 3 of NTP report 551 [11]. ^2^ Changes in comparison to the control value. The mean values were used for calculation. ^3^ Mean body weights and statistical results are taken from Table 11 of the NTP report 551 [11].

**Table 12 toxics-11-00991-t012:** Mean body weight of male rats from a 3-month gavage study.

Model	Goodness of Fit		BMD	BMDL	Rationale for Model Selection by BMDS Software
	*p*-Value	AIC			
Exponential 3 R	0.26	267.31	468	418	From all models that provided an adequate fit (*p* > 0.1), the unrestricted power model was selected based on the lowest AIC.
Exponential 5 R	0.49	267.12	430	358	
Hill R	0.35	267.53	447	390	
Polynominal Degree 3 R	0.16	268.30	484	429	
Power R,	0.17	268.25	477	430	
Linear (UnR)					
Polynominal Degree 2 UnR	0.41	267.32	428	353	
Power UnR	0.77	266.73	438	372	

R = restricted; UnR = unrestricted; AIC = Akaike Information Criterion; BMD = benchmark dose, BMDL = lower confidence limit of the benchmark dose, BMDS = Benchmark Dose Software.

**Table 13 toxics-11-00991-t013:** Results of benchmark dose modeling from several endpoints received from 3-month or 2-year gavage studies from mice and rats performed for the NTP program.

Endpoint	Species	Sex	Study	NOAEL or LOAEL (mg/kg b.w./day) ^1^	BMDL (mg/kg b.w./day)
Hepatocellular adenoma	Mouse	Male	2 years	LOAEL 75 ^1^	8 ^3^
Hepatocellular carcinoma	Mouse	Male	2 years	LOAEL 75 ^1^	18
Hepatocellular adenoma and carcinoma combined	Mouse	Male	2 years	LOAEL 75 ^1^	8 ^4^
Histiocytic sarcomas	Mouse	Female	2 years	NOAEL 150 ^2^	239
Thymoma	Rat	Male	2 years	NOAEL 150 ^2^	307
Mammary gland carcinoma	Rat	Male	2 years	NOAEL 150 ^2^	311
Atrophied olfactory epithelium	Rat	Female	3 months	LOAEL 37.5 ^2^	5
Reduction in body weight	Rat	Male	3 months	NOAEL 150 ^2^	372

^1^ The LOAEL values were set by the Committee for Veterinary Medicinal Products (CVMP) in a Maximum Residue Level Assessment [9]. ^2^ The NOAEL/LOAEL values were estimated by the authors of this article, as discussed in the respective results parts. ^3^ Suggested for use as POD (point of departure). ^4^ Calculated using the approach that includes skipping the high-dose group (300 mg/kg b.w./day).

**Table 14 toxics-11-00991-t014:** Comparison of the calculated ADI value of 16 µg/kg b.w./day with the estimated exposure levels of isoeugenol in Europe and in the USA.

Calculated ADI Value: 16 µg/kg b.w./day (8000 µg/kg b.w./day Divided by an Uncertainty Factor of 500)			
Exposure from food intake		Total exposure (food intake + consumer products)	
Europe	USA	Europe	USA
1.95 µg/kg b.w./day	0.72 µg/kg b.w./day	2.35 µg/kg b.w./day	1.12 µg/kg b.w./day

## Data Availability

Publicly available datasets were analyzed in this article. New derivative data presented in this article are available in the Appendix A.

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
