# Peer review of "Risk Assessment of Isoeugenol in Food Based on Benchmark Dose—Response Modeling"

_toxics, 2023, doi:10.3390/toxics11120991_

Round 1

Reviewer 1 Report

Comments and Suggestions for Authors

This is an excellent paper which gives the reader an overview of risk assessment in food safety, with an accurate description of data management and indexes elaboration.

The readability is very good and the references section is complete and useful for the reader.

Minor revision needed.

The Introduction should be simplified. Indeed, many sections (i.e., lines 48-68) seem not necessary to introduce this work. This is not a review or a guideline, so, the introduction should be only focused on the topic of this study, without explaining the meaning of each parameter, index, etc. of risk assessment.

Please improve readability and check some typos at lines 77-78, 92-93, 102, 160, 383-384, 421, 605.

Line 227: the meaning of such percentage is not well clear. Please improve this part.

Is a reference available at lines 317-318?

Please add a reference to related table at lines 345-347

Line 414: is 325 ± 3 the right value? Please verify.

Line 472: Is a legal limit available?

Lines 515-516: please add these regulations in references section.

Line 536: more comments needed about MoE values lower than 10000. Please improve the discussion.

Lines 563 and 570: 0.3

Line 609: 3400? Please verify.

Lines 621-649: This last section of conclusion should be simplified, substantially. It is suggested moving comments to discussion. Moreover, please do not use references in Conclusion.

Line 628: the explanation of the meaning of such acronym is needed.

Reference n.47: Please modify the author: European Commission

Tables 4, 8, 10: Please modify the last column: “Of the …” is a strange way to start a description. Please modify.

Tables 3-4: please explain the meaning of acronyms using footnotes

Tables 2, 7 and 9 (footnotes): Please report the reference [4] only once.

Figures 2-3 (caption): please delete underlining

Comments on the Quality of English Language

Please improve readability and check some typos at lines 77-78, 92-93, 102, 160, 383-384, 421, 605.

Author Response

This is an excellent paper which gives the reader an overview of risk assessment in food safety, with an accurate description of data management and indexes elaboration.

The readability is very good and the references section is complete and useful for the reader.

Minor revision needed.

The Introduction should be simplified. Indeed, many sections (i.e., lines 48-68) seem not necessary to introduce this work. This is not a review or a guideline, so, the introduction should be only focused on the topic of this study, without explaining the meaning of each parameter, index, etc. of risk assessment.

RESPONSE: Thank you for the excellent suggestions to improve our paper. Due to major revisions, the sections in the introduction and the discussion were partly replaced and rewritten. The calculation of the MOE was omitted from the manuscript according to the suggestions from the other reviewers.

To improve the logical structure of the manuscript we moved the section which deals with the exposure and the regulatory status of isoeugenol to the introduction part. Conversely, we moved the explanations of genotoxicity together with the listing of the performed genotoxic studies to the discussion part.
Furthermore, a new calculation was added to the results part (BMD modeling for the endpoint of hepatocellular adenoma and carcinoma combined), which, however, does not influence the overall derived BMDL value.
Due to these extensive changes it was impossible to use track changes to visualize the amended parts in the manuscript. Hence, we responded to your comments by giving the new lines where the respective topic can be found in the revised manuscript.

Please improve readability and check some typos at lines 77-78, 92-93, 102, 160, 383-384, 421, 605.

RESPONSE:

Old: Lines 77-78 (old sentence)

However, it should be pointed out that at low concentrations thresholds for direct carcinogens could be detected.

Revised sentence in the revised manuscript (New)

However, it is now increasingly accepted that thresholds for direct carcinogens could be detected at low concentrations.

Old: 92 – 93

Hence, to establish a scientifically valid risk assessment for isoeugenol, knowledge about the mechanism is needed how isoeugenol may induce cancer.

New:

Hence, to establish a scientifically valid risk assessment for isoeugenol, it is important to know if it acts via a genotoxic or a non-genotoxic mechanism.

Old: 102

From these 10 key characteristics the studies that are summarized under the key characteristic “is genotoxic” are important to decide if an Acceptable Daily Intake (ADI) value can be used or an MOE value has to be calculated.

New

To decide if an Acceptable Daily Intake (ADI) value or an MOE value has to be used for risk assessment, the key characteristic “is genotoxic” is of particular importance.

Old:160

The data that was used for BMD modeling is published in the NTP Technical Report No. 551 ‘Toxicology and carcinogenesis studies of isoeugenol in F344/N rats and B6C3F1 mice’.

New:

The data used for BMD modeling was taken from the NTP Technical Report No. 551 ‘Toxicology and carcinogenesis studies of isoeugenol in F344/N rats and B6C3F1 mice’.

Old: 383-384

The authors of this article are of the opinion that for the endpoint of an atrophied olfactory epithelium the observed cases at the low dose group can be considered as an adverse effect.

New:

The obtained BMDL value of 5 mg/kg b.w./day is below the low dose level of 37.5 mg/kg b.w./day. For this endpoint (atrophied olfactory epithelium) the low dose level can be taken as the LOAEL, as the authors of this article believe that the observed incidences at the low dose level (30 % for the male rats and 10 % for the female rats) are a significant increase in comparison to the control group were no rats with an atrophied epithelium were noted in the 3 months rat gavage study.

Old: 421

Several viable models were obtained using this approach (excluding the two low dose levels). The software recommended the unrestricted power model (BMD = 438 mg/kg b.w./day; BMDL = 372 mg/kg b.w./day) due to the lowest AIC value (Table 10 and Figure 9).

New:

Running a BMDS analysis without the 2 low dose levels revealed several viable models and the software recommended the unrestricted power model (BMD = 438 mg/kg b.w./day; BMDL = 372 mg/kg b.w./day) due to the lowest AIC value (Table 10 and Figure 9).

Old: 605

The BMD-based approach instead of the LOAEL-based approach for risk assessment of isoeugenol led to a reduction of the POD value from 75 µg/kg b.w./day to 8 µg/kg b.w./day.

New

The BMD-based approach revealed a POD of 8 µg/kg b.w./day. The POD of 8 mg/kg b.w./day was significantly below the LOAEL value of 75 µg/kg b.w./day, which was used as POD by the CVMP for the calculation of the actual ADI value. The use of the new BMD-based POD led to a significant reduction of the ADI value.

Line 227: the meaning of such percentage is not well clear. Please improve this part:

Sentence in Line 227:

A clear dose-response relationship is not visible. The incidence persisted between 33% and 37% at the investigated dose levels

RESPONSE:

New sentence:

The incidences at the low, the intermediate and the high dose level were between 66% and 74% for the endpoint of hepatocellular adenoma, between 36% and 38% for the endpoint of hepatocellular carcinoma and consistently at 86 % for the endpoint of hepatocellular adenoma and carcinoma combined.

Is a reference available at lines 317-318?

Sentence in lines 317-318

However, due to the statistically significant positive dose-related trend, the authors set the NOAEL value at the intermediate dose level of 150 mg/kg b.w./day.

RESPONSE

The sentence in lines 317-318 (unrevised manuscript) is a statement by the authors of this manuscript. To clarify this, we have supplemented the sentence as follows:

However, due to the statistically significant positive dose-related trend, we consider a NOAEL value at the intermediate dose level of 150 mg/kg b.w./day to be suitable.

Please add a reference to related table at lines 345-347

RESPONSE:

The data in the sentence refers to table 6. We have added this reference to the respective sentence (line 312 of the revised manuscript).

Line 414: is 325 ± 3 the right value? Please verify.

RESPONSE:

We are sorry, this was a typo. The value in table no. 11 (Mean body weights from male mice and rats from 3 months gavage studies) (325 ± 3) is correct. We have corrected the error in the text.

Line 472: Is a legal limit available?

RESPONSE:

The value of the legal limit and its calculation is in detail described in section 4.4 ‘Re-calculation of the MRL value for fin fish’. We have added a reference to the respective line.

Lines 515-516: please add these regulations in references section.

RESPONSE:

We have moved this section from the conclusion into the results section (4.2 ‘Risk assessment of isoeugenol’). In addition, we have now cited the original publications of the studies instead of the OECD guidelines.

Line 536: more comments needed about MoE values lower than 10000. Please improve the discussion.

RESPONSE:

Section 4.3 ‘Calculation of MOE value’ was skipped from the manuscript and, thus, also the sentence in line 536 about the MOE results.

Lines 563 and 570: 0.3

RESPONSE: We have changed the weight information from 0.30 kg to 0.3 kg

Line 609: 3400? Please verify.

RESPONSE: As the calculation of the MOE value was skipped, this part of the conclusion was also skipped.

Lines 621-649: This last section of conclusion should be simplified, substantially. It is suggested moving comments to discussion. Moreover, please do not use references in Conclusion.

RESPONSE:
Lines 621 to 634 were moved to section 4.2 of the discussion ‘Risk assessment of isoeugenol’.
Lines 635 to 649 were skipped.

Line 628: the explanation of the meaning of such acronym is needed.

RESPONSE: OECD (Organization for Economic Co-operation and Development).
We have additionally added a table of abbreviations to the end of the manuscript.

Reference n.47: Please modify the author: European Commission

RESPONSE: We added ‘European Commission’ to the reference.

Tables 4, 8, 10: Please modify the last column: “Of the …” is a strange way to start a description. Please modify.

RESPONSE: We changed the term ‘Of the models..’ to ‘From all models that provided an adequate fit.’

Tables 3-4: please explain the meaning of acronyms using footnotes

RESPONSE: We added the meanings for the abbreviations of AIC, BMD, BMDL and BMDS to the footnotes of tables 4 and 5.

Tables 2, 7 and 9 (footnotes): Please report the reference [4] only once.

RESPONSE: We have deleted the duplicate references

Figures 2-3 (caption): please delete underlining

RESPONSE: We have deleted the lines under the words ‘unrestricted’ and ‘restricted’.

Comments on the Quality of English Language

Please improve readability and check some typos at lines 77-78, 92-93, 102, 160, 383-384, 421, 605.

RESPONSE: We are of the opinion that with the above described revisions the English quality as well as the readability was improved. The typos were corrected.

Reviewer 2 Report

Comments and Suggestions for Authors

In the present manuscript, the authors utilized BMD analysis approach vs LOAEL/NOAEL approach to derive POD for supporting risk assessment of isoeugenol in food.  Several major concerns are listed below:

Genotoxicity and risk assessment approach: The available in vitro and in vivo data provide sufficient weight of evidence that isoeugenol is non-genotoxic and a threshold-based approach should be considered for risk assessment. The inclusion of the MOE approach is confusing and misleading. This makes the authors’ statement contradictory to each other (eg. The last two sentences in the abstract). Despite that a 3.2-fold increase in the frequencies of micronucleated erythrocytes in female mice in the highest dose-group of 600 mg/kg bw/day was observed in the NTP a peripheral blood micronucleus test, in another two in vivo micronucleus tests in male and female mice, negative results were obtained at even higher dose level. In addition, no evidence of genotoxic potential could be observed in the rat in vivo UDS test in male and female rats administered doses of up to 2000 mg/kg (males) and 1250 mg/kg (females).  These data were included in both the CVMP document and in the RIFM assessment to serve as critical WOE for the non-genotoxic conclusion but were missed in the manuscript.  The authors should take all available data into consideration for the genotoxicity discussion and for the risk assessment approach selection.

BMD modeling:  Reevaluation is needed for the BMD analysis on critical endpoints. For cancer endpoint BMD analysis, the combined adenoma and carcinoma is recommended. Viable model can be obtained if you drop high dose group per EPA BMDS technical guidance.  When modeling adenoma and carcinoma separately, the same approach (drop the high dose group) can be taken since the high dose group did not follow dose response and the BMR is close to or lower than the lowest dose group.

ADI derivation: The authors followed the CVMP approach of adding an UF of 5 to account for the potential seriousness and irreversibility of the effect combined with the deficiencies of the study for ADI derivation. This UF is often considered when using LOAEL as POD and is associated with the LOAEL to NOAEL extrapolation. It is not applicable for BMDL approach since the BMD model already take the potential seriousness of the effects into consideration (the shape of the curve).

 In the Conclusion section (3rd paragraph), the authors mentioned about risk management measures and suggested similar measures for isoeugenol as methyleugenol. It should be noticed that there is sufficient evidence to support that methyleugenol is genotoxic (IARC)--- It is known that Methyl eugenol induces DNA adducts formation, but isoeugenol does not interactive with DNA. Different risk assessment approach and risk management measures should be considered based on the different MOA. Again, for the genotoxicity discussion and suggestion in the last two paragraphs should be revised based on all available genotoxicity data (see #1).

Comments on the Quality of English Language

There is no major concern on the English writing.  Several minor revisions are needed.

“However” is used to introduce a statement that contrasts with or seems to contradict something that has been said previously. The authors used “however” in many places that the following statement does not contradict to the previous statement.  Proper wording is needed to replace "however". For example: Line 66, Line 77, Line 80, Line 96, Line 138.

Line 93, the mechanism is needed (on) how isoeugenol may induce cancer.

Line 105: “These studies that predict a direct genotoxic and consequently non-thresholded mechanism revealed negative results.” This sentence is confusing and needs to revise.  Suggest to spilt into two sentences. To make it clear, only positive results from these tests support a mutagenic potential, the results for isoeugenol are all negative.

Author Response

In the present manuscript, the authors utilized BMD analysis approach vs LOAEL/NOAEL approach to derive POD for supporting risk assessment of isoeugenol in food.  Several major concerns are listed below:

Genotoxicity and risk assessment approach: The available in vitro and in vivo data provide sufficient weight of evidence that isoeugenol is non-genotoxic and a threshold-based approach should be considered for risk assessment. The inclusion of the MOE approach is confusing and misleading. This makes the authors’ statement contradictory to each other (eg. The last two sentences in the abstract). Despite that a 3.2-fold increase in the frequencies of micronucleated erythrocytes in female mice in the highest dose-group of 600 mg/kg bw/day was observed in the NTP a peripheral blood micronucleus test, in another two in vivo micronucleus tests in male and female mice, negative results were obtained at even higher dose level. In addition, no evidence of genotoxic potential could be observed in the rat in vivo UDS test in male and female rats administered doses of up to 2000 mg/kg (males) and 1250 mg/kg (females).  These data were included in both the CVMP document and in the RIFM assessment to serve as critical WOE for the non-genotoxic conclusion but were missed in the manuscript.  The authors should take all available data into consideration for the genotoxicity discussion and for the risk assessment approach selection.

RESPONSE: We agree with the opinion of the CVMP in the MRL report that there is no concern for genotoxicity of isoeugenol. Hence, we omitted the calculation of the MOE value from the manuscript.

BMD modeling:  Reevaluation is needed for the BMD analysis on critical endpoints. For cancer endpoint BMD analysis, the combined adenoma and carcinoma is recommended. Viable model can be obtained if you drop high dose group per EPA BMDS technical guidance.  When modeling adenoma and carcinoma separately, the same approach (drop the high dose group) can be taken since the high dose group did not follow dose response and the BMR is close to or lower than the lowest dose group.

RESPONSE:

According to section 2.3.6 of the Technical Guidance it is possible to omit the highest dose group if none of the available models provide an adequate fit (according to an objective criterion like p < 0.10 for a goodness of fit test). This approach should not be used to refine an already adequate fit (as we have for the adenoma and carcinoma alone). Hence, this approach cannot be used for the separated evaluations of carcinoma and adenoma, where models with an adequate fit are available without dropping dose groups.

However, no viable model was received for the endpoint of adenoma and carcinoma combined when using all dose groups. Hence, the approach to skip the highest dose group was in accordance with the guideline for this endpoint. The guideline further recommended a clear justification for dropping of dose groups. This justification is given by the fact that the incidence in the low, the intermediate and the high dose group is the same (43 %, no dose-response relationship between the dose groups) and the BMDL value is near the low dose group.

We applied this approach (dropping of the high dose group) for the endpoint of carcinoma and adenoma combined and received several models with an adequate fit (p > 0.1). The BMDL value received has the same amount (8 mg / kg) as the BMDL value for the endpoint of hepatocellular adenoma alone when using all dose groups. Hence, the additional calculation using the endpoint of adenoma and carcinoma combined does not change the resulting BMDL value which remains at 8 mg/kg b.w./day.

ADI derivation: The authors followed the CVMP approach of adding an UF of 5 to account for the potential seriousness and irreversibility of the effect combined with the deficiencies of the study for ADI derivation. This UF is often considered when using LOAEL as POD and is associated with the LOAEL to NOAEL extrapolation. It is not applicable for BMDL approach since the BMD model already take the potential seriousness of the effects into consideration (the shape of the curve).

RESPONSE:

In our answer we refer to the Scientific opinion ‘Guidance on selected default values to be used by the EFSA Scientific Committee, Scientific Panels  and Units  in the absence of actual measured data’ from the EFSA Scientific Committee (EFSA Journal 2012, 10 (3): 2579):

It is correct that the UF which considers the use of a LOAEL instead of a NOAEL can be skipped, if a BMDL value is used.

According to the EFSA guideline mentioned above the UF for a LOAEL should be determined on a case by case basis. In the MRL assessment report the CVMP used a UF of two to consider the use of a LOAEL instead of a NOAEL. As we used a BMDL value we skipped this UF factor of 2 for our ADI calculation.

However, to our understanding the UF value of 5 considers two other uncertainties. In the guidelines these uncertainties are designated as study deficiencies and severities (seriousness) of the observed effect. Both UF values must be determined on a case by case basis and for or both uncertainties the CVMP set a value of 5 in the MRL report for isoeugenol. These value of 5 is independent from the use of a LOAEL and hence, was not skipped in our ADI calculation.

In the Conclusion section (3rd paragraph), the authors mentioned about risk management measures and suggested similar measures for isoeugenol as methyleugenol. It should be noticed that there is sufficient evidence to support that methyleugenol is genotoxic (IARC)--- It is known that Methyl eugenol induces DNA adducts formation, but isoeugenol does not interactive with DNA. Different risk assessment approach and risk management measures should be considered based on the different MOA. Again, for the genotoxicity discussion and suggestion in the last two paragraphs should be revised based on all available genotoxicity data (see #1).

RESPONSE:
We have skipped methyleugenol from the manuscript.

Comments on the Quality of English Language

There is no major concern on the English writing.  Several minor revisions are needed.

“However” is used to introduce a statement that contrasts with or seems to contradict something that has been said previously. The authors used “however” in many places that the following statement does not contradict to the previous statement.  Proper wording is needed to replace "however". For example: Line 66, Line 77, Line 80, Line 96, Line 138.

RESPONSE: The language was revised and the use of „however“ checked and replaced where inappropriate.

Line 93, the mechanism is needed (on) how isoeugenol may induce cancer.

RESPONSE:
In the end end of section 4.2 ‘Risk assessment of isoeugenol’ we agree with the CVMP committee that isoeugenol is not genotoxic and that the carcinogenic activity of isoeugenol is most likely due to non genotoxic mechanism.

Line 105: “These studies that predict a direct genotoxic and consequently non-thresholded mechanism revealed negative results.” This sentence is confusing and needs to revise.  Suggest to spilt into two sentences. To make it clear, only positive results from these tests support a mutagenic potential, the results for isoeugenol are all negative.

RESPONSE: We have revised the sentences as follows:

Substances that are positive in these mutagenicity studies could be considered as a direct genotoxic substance without a threshold. However, the results for isoeugenol in these mutagenicity studies were all negative.

Reviewer 3 Report

Comments and Suggestions for Authors

Dear Authors,

Some issues need to be corrected, as listed below;

Lines 103-157; The names (numbers) of the cell lines studied should be given in detail, and also for biological studies performed the information in the each case; was it in vitro or in vivo study (studied organism model; rat, turkey, mice, zebrafish or others).

Also concentrations used in the studies should be given (line 144). Also more detail for studies described in the line 222 (including references for the study).

It would be helpful for readers to have in the manuscript the short abbreviation list because many shortcuts of the institutions and the protocols are used in the text.

Line 155-156 and in the Abstract line 13; it should be ; Benchmark Dose (BMD). Similarly; line 173; Benchmark Response Factor (BMRF)

What is EPA (see line 177).

Lines 155-157- citations needed.

References section;

Citations should be revised and volume (s) number should be given (see ref. 7, 15, 23, 25, 27, 28, 45)

The Journal’s abbreviations should be properly applied (see e.g. ref. 45, line 779)

In vitro, in vivo (lines;142,  686, 734, and more), organisms (e.g. Drosophila melanogaster – see line 104, 723, ) and plant systematic names – should be in italics.

Author Response

Dear Authors,

Some issues need to be corrected, as listed below;

Lines 103-157; The names (numbers) of the cell lines studied should be given in detail, and also for biological studies performed the information in the each case; was it in vitro or in vivo study (studied organism model; rat, turkey, mice, zebrafish or others).

RESPONSE: The corresponding section has been revised and additional information about the cited studies (e.g. cell names) was added.

Also concentrations used in the studies should be given (line 144). Also more detail for studies described in the line 222 (including references for the study).

RESPONSE: Line 144: We have added further information (e.g. the used cell line) to this section (key characteristic ‘induces oxidative stress’, now in section 4.2). However, as cell lines were used in this study, a comparison with the mice and rat gavage studies is not completely possible. Due to this reason, we did not mention the concentrations that were used in the cell assays. The concentrations can easily be found in the cited publications.

Line 222: We have added the concentrations of isoeugenol that were used in the cited study, as well as the source of the study (NTP Technical Report No. 551).

It would be helpful for readers to have in the manuscript the short abbreviation list because many shortcuts of the institutions and the protocols are used in the text.

RESPONSE: We have added a table with the abbreviations to the end of the manuscript.

Line 155-156 and in the Abstract line 13; it should be ; Benchmark Dose (BMD). Similarly; line 173; Benchmark Response Factor (BMRF)

RESPONSE: We have changed the spelling accordingly.

What is EPA (see line 177).

RESPONSE: EPA is the abbreviation for Environmental Protection Agency. Explanation of abbreviation was introduced.

Lines 155-157- citations needed.

Sentence: For the calculation of the necessary Point of Departure (POD) value, the benchmark dose (BMD) approach was used, which is nowadays recommended instead of the NOAEL approach.

RESPONSE:  We added a reference with the number 12 (EFSA Journal 2022: Guidance on the use of the benchmark dose approach in risk assessment) to this sentence.

References section;

Citations should be revised and volume (s) number should be given (see ref. 7, 15, 23, 25, 27, 28, 45)

RESPONSE: We have added the Volume number to each reference (please be aware that the reference numbers for the cited papers often changed during the revision of the manuscript).

The Journal’s abbreviations should be properly applied (see e.g. ref. 45, line 779)

RESPONSE: We have abbreviated the name of the journal from ‘Food Technology’ into ‘Food Technol.’ Please be aware that in the revised manuscript this citation has the reference number 48.

In vitro, in vivo (lines;142,  686, 734, and more), organisms (e.g. Drosophila melanogaster – see line 104, 723, ) and plant systematic names – should be in italics.

RESPONSE: We have italicized the relevant words. Note that MPDI style guide does NOT suggest in vivo/vitro in italics. We have made the corrections to the style guide.

Round 2

Reviewer 2 Report

Comments and Suggestions for Authors

Thanks for revising the manuscript and several minor edits can be made before the final publication.

1. Table 13, the BMDL from the combined adenoma and carcinoma can be included in this table to support your overall POD conclusion. 

2. Since genotoxicity is most important for the threshold-based risk assessment decision making, recommend removing the non-relevant KCC discussion on "is electrophilic" and "induce oxidative stress" (from line 785-line 806) to avoid confusion.

3. page 27, line 839, "a non-threshold mechanism can be assumed" should be revised to "a threshold-based mechanism can be assumed"

Author Response

Thanks for revising the manuscript and several minor edits can be made before the final publication.

1. Table 13, the BMDL from the combined adenoma and carcinoma can be included in this table to support your overall POD conclusion. 

RESPONSE: The BMDL was included and a footnote for explanation added to Table 13.

2. Since genotoxicity is most important for the threshold-based risk assessment decision making, recommend removing the non-relevant KCC discussion on "is electrophilic" and "induce oxidative stress" (from line 785-line 806) to avoid confusion.

RESPONSE: The authors agree and the lines were deleted as requested.

3. page 27, line 839, "a non-threshold mechanism can be assumed" should be revised to "a threshold-based mechanism can be assumed"

RESPONSE: Thank you for spotting this omission. The sentence was changed.